# Miniaturized Antenna Array-Based Novel Metamaterial Technology for Reconfigurable MIMO Systems

**DOI:** 10.3390/s23135871

**Published:** 2023-06-25

**Authors:** Humam Hussein, Ferhat Atasoy, Taha A. Elwi

**Affiliations:** 1Department of Computer Engineering, Karabuk University, Karabuk 78050, Turkey; eng.humam94@gmail.com (H.H.); ferhat.atasoy@gmail.com (F.A.); 2International Applied and Theoretical Research Center (IATRC), Baghdad 10001, Iraq

**Keywords:** 5G, MIMO, sub-6, MTM, reconfigurable, BER

## Abstract

In this work, a highly miniaturized microstrip antenna array based on two elements is proposed for multiple inputs multiple outputs (MIMO) application systems at sub-6 GHz frequency bands. The antenna is structured from a meander line in conjugate with an interdigital capacitor when excited through the monopole basic antenna. The proposed antenna elements are separated with a Minkowski factor-shaped metamaterial (MTM) column to achieve a separation distance (D) of 0.08λ at 3 GHz when printed on an FR-4 substrate. Later on, the antenna performance in terms of bandwidth and gain is controlled using a photonic process based on optical active switches based on light-dependent resistances (LDR). Therefore, the reconfiguration complexity with such a technique can be eliminated significantly without the need for a biasing circuit. The antenna design was conducted through several parametric studies to arrive at the optimal design that realizes the frequency bandwidth between 3 and 5.5 GHz with a maximum gain of about 4.5 dBi when all LDR terminals are off. For a wireless channel performance study-based massive MIMO environment, the proposed antenna is suitable to be configured in arrays of 64 × 64 elements. From this study, it was found the maximum bit error rate (BER) does not exceed 0.15 with a channel capacity (CC) of 2 Gbps. For validation, the antenna was fabricated based on two elements and tested experimentally. Finally, it was revealed that the measured results agree very well with simulations after comparing the theoretical calculations with the measured data.

## 1. Introduction

Recently, microstrip antennas were introduced in many wireless communication networks including 5G systems for their desired properties, including a low profile and inexpensive fabrication costs [1]. Nevertheless, MTM and specific composite right/left-handed (CRLH) structures are introduced to the microstrip antenna designs from different research groups to enhance their limitations including low gain bandwidth products [2]. Therefore, many researchers applied their scientific research and theoretical studies to develop different microstrip antennas for different applications. One of the most popular applications of CRLH-based microstrip antennas is the MIMO antenna array [3]. This technology received high attention due to its unique specifications, among many others, its high channel capacity with minimum latency [3]. For this, the authors in [4] developed an array of two printed transparent circuit antennas with low gain and limited bandwidth for MIMO applications at sub-6 GHz. In [5], a design of four elements was sequentially fed and mounted on a substrate backed with an electromagnetic bandgap structure of sub-6 GHz bands. The antenna’s orientation was introduced to avoid a high coupling effects due to skew waves. However, the proposed MTM was presented to minimize the surface wave’s effects from the antenna back panel. Other researchers developed, for example, a low-profile antenna in [2] based on antenna elements of Hilbert-shaped fractal metamaterials to form a 2D-rectangular array fed with a coplanar waveguide feed line. Another design was proposed for sub-6 GHz applications based on a wideband elliptically slotted semi-circular patch element with a defected ground plane structure [4]. In [5], efforts were applied to design a directive pointer antenna for sub-6 GHz systems including 5G applications. Another work was published in [6] to realize a design of MIMO antenna based 2 elements at sub-6GHz frequency bands for portable devices. A dual frequency band reconfigurable antenna-based portable device was developed for 5G systems in [7]. A dual band antenna based on compact flexible structure for GSM and Wi-Fi applications [8]. A design of an antenna is bansed MTM structure for reconfigurable systems at sub-6GHz bands [9]. A MIMO antenna array system was proposed based on a pair of transparent antenna elements at sub-6 GHz frequency bands [10]. In [11], a metasurface reflector with a four-port wideband MIMO antenna array systems was developed to support 5G sub-6 GHz applications.

Later, several studies were applied to design an antenna that supports more than one application; for instance, in [12] an antenna was developed for 5G MIMO systems and the Internet of Things. In [13], a dual-band antenna based on composite right-and left-handed transmission lines was developed for MIMO systems at sub-6GHz bands. Additionally, in [14], a design was introduced for the applications of 5G MIMO antenna arrays to be applied in smartphones at the frequencies in the ranges of 3.41–3.63 and 4.82–5.23 GHz. In [15], two wideband PIFA elements were offered as a possible solution for constructing a miniaturized MIMO array to suit 5G mobile handsets [15]. A design of a compact broadband bandstop filter based on composite right/left handed transmission line was proposed to reduce the mutual coupling between the antenna elements in MIMO systems at 3.5GHz bands [16]. In the same manner, a design was suggested in [17] using a split ring resonators to reduce the mutual coupling between antenna at sub-6 GHz for 5G MIMO systems. A reconfigurable antenna element was designed to provide a wide and narrow frequency band for smart MIMO antenna arrays [18]. A design of an MTM-based ultra-wideband antenna was developed to form a MIMO array [19]. A design of a solar panel integrated 3D MIMO antenna array for 5G self-powered communication systems was proposed in [20]. For frequencies between 3.4 and 3.8 GHz, a MIMO antenna array was developed with a partial ground plane [21]. A wide-band antenna array based on MTM was invented to cover the frequency band in 5G systems [22]. Another design was proposed based on flexible transparent wideband antenna elements to form a MIMO antenna array to operate at sub-6 GHz [23]. An electromagnetic band gap structure was introduced between antenna elements to diminish the effects of mutual coupling among antennas in a single array form to suit the applications of 5G systems [24]. A meander line-based antenna array was developed for MIMO applications by separating the antenna elements using an electromagnetic band gap structure at sub-6 GHz bands [25].

## 2. Antenna Array Design Details

This antenna array is designed from two antenna elements printed on the same substrate for MIMO system design. Each antenna element is structured from three main parts: a printed monopole, an interdigital capacitor array, and a meander line. The proposed antenna is fed with CPW to eliminate the ground plane on the same patch side as well as ensure harmonics generation by accumulating the electrical charge on the ground plane [12]. Therefore, the proposed meander line is fetched to the ground plane of the proposed CPW structure. In such a structure, the effects of field gradients can be enhanced altimetry to realize bandwidth enhancements [3]. For this, the antenna gain bandwidth would be enhanced to realize the desired frequency resonances with maximum size reduction [5]. Nevertheless, the proposed antenna performance can be controlled by switching the connections between the antenna meander line and the ground plane through four LDR resistors. The proposed antenna array occupies an area of 50 × 30 mm^2^ when printed on an FR4 substrate of 1 mm thickness.

For the MIMO configuration, the authors applied the design of two elements that are mounted on the same substrate with the same direction, as shown in Figure 1. The antenna array is defective with an MTM structure at the center position between the antenna elements to ensure surface wave sprucing at the desired frequency bands [6]. The proposed MTM structure is constructed using Minkowski fractal design with the third iteration. In such a structure, a high capacitive load is coupled near the edges of the proposed monopoles; in it, most fringing effects are initiated [3]. Logically, positioning such an MTM at the center of the proposed array would be an excellent option. In another aspect, the proposed MTM is structured to provide a single negative performance in which a high coupling redaction can be achieved [7]. Therefore, the surface wave mitigation between the antenna elements would vanish smoothly without significant phase reflections [11]. Simultaneously, a high diversity can be achieved by directing most antenna radiation away from the array center through the inductive imposing effect from the proposed MTM fractal structure [14].

It is important to note that the introduction of the meander line has the effect of producing a frequency band that is distinct from the band emitted by the monopole, as will be demonstrated later. To regulate the mobility of the antenna surface current, an LDR switch is used to link the planned meander line to the antenna ground plane. The key benefit of including the interdigital capacitor in the design is that it may be used to balance the monopole antenna structure’s induction effect and match the electromagnetic aperture coupling to the free space impedance. As will be demonstrated later [8], this would considerably increase the antenna radiation efficiency. The suggested antenna is then printed on an RF-4 substrate that is 1 mm thick. Ultimately, an RF-4 substrate with a 1 mm thickness and 4.4 dielectric constants is used to print the suggested antenna. Table 1 contains the remaining geometrical information about the proposed array.

## 3. MTM Analysis and Discussion

As shown in Figure 1, the suggested MTM structure is built from a 1D array with four unit cells (b). The third iteration of the Minkowski fractal describes the form of each individual unit cell. A circuit model is developed to describe the performance based on a transmission line theory to analyze the suggested unit cell features. In essence, a circuit model with a mix of right- and left-hand branches may be created from such a unit cell, as illustrated in Figure 2a.

In terms of mathematics, the suggested unit cell was examined from various orientation angles at normal and tangential electric field incidences. Yet, in this inquiry, the planned unit cell iteration was raised from the first to the fourth with a single step. This was done to make sure that the performance of the suggested unit cell is improved by the iteration effects. As a result, the suggested unit cell is situated inside a fictitious waveguide to track the S-parameter spectra at the targeted frequency range. With the application of two electrical and two magnetic barriers to the virtual waveguide, the effective medium theory is invoked in this simulation process. A similar procedure is used to simulate the creation of transverse electromagnetic modes as a plane wave [10].

Now, after inserting the proposed unit cell with different orientations and different iterations, the calculated S-parameters, in terms of S_12_ and S_11_ spectra, are shown in Figure 2b. The suggested unit cell was discovered to offer a cut-off frequency for the frequency bands from 3 up to 5.3 GHz. The proposed unit cell performance in terms of permittivity and permeability is presented in Figure 2c. It was found that the proposed unit cell realizes multiple bands with −εr and −µr within the frequency band of interest. Such an observation realizes that the unit cell is an excellent candidate for mutual coupling reduction to suite MIMO applications; that is very useful for size reduction [5]. Next, as a result, the suggested unit cell was nominated as an outstanding option for coupling reduction in MIMO array systems as a result of this accomplishment. The comparable circuit model based on the developed RLC network, which is often built as an interdigital capacitor (IDC) in series connected to a compact series inductor (CSI) and Hilbert fractal, is used to analytically determine the suggested sensor design based on an equivalent circuit model. The lumped elements Richard model [10] was used to create the suggested structure equivalent circuit model. A 50 input impedance RF source is coupled in series with a R-L-C parallel branch to form the suggested circuit architecture, as seen in Figure 2. An inductive segment LT and capacitive air gaps Cgap, which were previously shown in Figure 2, set apart the primary transmission line. The recommended S-parameters for the circuit model were reviewed and contrasted with those obtained using CST MWS, considering the lumped components indicated that were simulated in Advanced Devices Simulator (ADS). Table 2 contains a list of the assessed RLC parts.

### 3.1. Design Methodology

#### 3.1.1. Monopole Antenna Performance

In this section, the proposed monopole element performance in terms S_11_ spectra are investigated with respect to varying the separation distance between the monopole and the ground plane (Xg). Therefore, the considered distance Xg was changed from 1 to 5 mm with a step of 1 mm with respect to monitoring S_11_ spectra as shown in Figure 3. It was observed from the obtained results that the antenna provides two frequency bands around 1.9 and 4.5 GHz. Nevertheless, it is seen that the proposed antenna matching impedance is insignificantly changed with changing Xg at the first band; however, at the second band, the matching impedance is improved pointedly at Xg = 3 mm; then, it returns back to being degraded due to increasing the distance. Additionally, it was observed that the frequency resonance location is not changed due to the fact of keeping the monopole length fixed to a certain length that has no effect on the current motion [12]. Moreover, it was observed that the generated frequency resonances are relative to the multiple constants of the halves of the wavelengths, which depends on the antenna’s electrical lengths’ harmonics [13]. Consequently, the length of the proposed monopole was found to be relative to the physical length after the proposed CPW feeding structure. Such an observation reveals that the capacitive effects between the antenna ground plane and the monopole structure could accumulate the electrical charges significantly to increase the antenna input impedance over the microwave source [15]. For this, the authors considered Xg = 2 + 1 mm to be the best choice for the next design step; this is because the matching was found to be around −35 dB at 4.35 GHz.

Now, to investigate the effects of increasing the ground plane length on the antenna frequency resonance, the authors considered Xg = 3 mm, but the ground plane length was changed from 8 to 5 mm with a step of 1 mm. It was observed that with decreasing the ground plane length, there was a significant decay in the antenna second mode frequency resonance. Therefore, it is concluded the ground plane length (Yg) is better to be 5 mm for our applications. Reducing the length to less than 5 mm is not very desired during the fabrication process due to soldering limitations [4]. The calculated results are shown in Figure 4 in terms of S_11_ spectra with respect to varying the ground plane length (Yg).

For further analysis, the authors studied the effects of varying the monopole width on the antenna S_11_ spectra. Therefore, the monopole width (Xm) was changed from 1 to 5 mm. It was found that the observable variation in the term of S_11_ spectra is specific to bandwidth, matching, and frequency resonance, as depicted in Figure 5. This is attributed to the fact that current motion is usually on the antenna length that follows the dimension with a larger conduction area [12]. For this, such phenomena are well known from different previous publications that discussed the effects of current motion on antennas’ maximum dimension [12].

#### 3.1.2. Interdigital Capacitor Effects

Since the introduction of the IDC structure to the proposed antenna could cause a significant performance change, the authors applied a study to evaluate the antenna S_11_ and gain spectra. Therefore, the proposed IDC iteration number was changed from the first to the fourth degree. The effects of that on S_11_ and gain spectra are shown in Figure 6. We realized that the proposed antenna bandwidth of the second mode at 3.8 GHz is significantly affected. Another mode was found to be generated at 5.8 GHz due to the proposed IDC introduction as seen in Figure 6b. An approximate expression for the interdigital capacitance is given by [1].
(1)$$C=\frac{\varepsilon_{r}+1}{W^{\prime }}l\;\left[\left(N-3\right)A_{1}+A_{2}\right]$$
where *C* is the capacitance per unit length (*l*) with respect to the finger width (*W*), *A*_1_ (the interior) and *A*_2_ (the two exteriors) are the capacitances per unit length of the fingers, and *N* is the number of fingers and can be expressed in microns. For infinite substrate thickness (or no ground plane), *A*_1_ = 4.409 × 10^−6^ pF/mm and *A*_2_ = 9.92 × 10^−6^ pF/mm.

Now, the antenna gain spectra over the entire frequency band of interest are discussed with respect to the same IDC iteration variation as shown in Figure 6b. It was found that the proposed antenna gain is enhanced significantly by increasing the proposed IDC iteration increase. Such a fact is attributed to the effects of the proposed IDC on the inherent inductive behavior of the proposed monopole [15]. Thus, the introduction of the proposed IDC minimizes the stored energy by the induction part in the monopole [16].

#### 3.1.3. Mender Line Effects

The authors introduced a meander line to examine the effects of coupling between the ground plane and the antenna parts. Such coupling is due to the effects of the electrical field fringing from the antenna edges [12]. The effects of such fringing could realize a further loss in the antenna performance, specifically in terms of gain [13]. We decided to apply a parametric study to the proposed meander line by changing the turn number from one turn to five turns concerning S_11_ and gain spectra variation. As shown in Figure 7a, it was found that the proposed antenna operates over several frequency bands and they increased with increasing the turn number. Such an observation is due to the fact that the current path can be increased further on a meander line [16]. Therefore, the introduction of the proposed meander line was found to be an excellent candidate to realize a significant enhancement to the antenna size reduction.

Now, the antenna gain was found to increase when increasing the turn number, as seen in Figure 7b. This realizes a significant enhancement due to the reduction in storing losses. Such a reduction is attributed to the negative effects of the induction part by the mender line on the capacitive behavior of the ground plane concerning the other antenna parts. It is good to mention that the authors did not increase the turn number further, because the effects of increasing the turn number on the antenna performance become very slow after the third turn, and due to antenna size limitations.

#### 3.1.4. MIMO Array Performance

Now, the proposed antenna was designed to be configured as an array with two elements, as shown in Figure 1. Therefore, to specify the effects of coupling on the proposed antenna performance, the separation distance (D) between the antenna elements was changed from 10 to 25 mm. As seen in Figure 8a, the proposed antenna array S_11_ spectra are insignificantly affected by increasing D. This is a usual response because the fact of changing S_11_ spectra is relative to varying the antenna elements’ dimensions [12]. However, in the case of S_12_ spectra, the manner of the antenna mutual coupling variation is different; it is increased with decreasing the distance between the antenna elements, as shown in Figure 8b.

Our philosophy in this research was to reduce the antenna element size within a miniaturized array configuration by adding MTM defects. Such an addition can be implemented to minimize the separation distance ultimately to 6 mm with coupling effects in terms of S_12_ below −20 dB, as seen in Figure 9. Such a performance was compared to the antenna array with and without MTM defects. The fact of reducing the mutual coupling with low separation distance is that such MTM defects apply a high surface impedance by producing a negative dielectric constant (−ε_r_) over the frequency bands of interest. Therefore, the propagated surface current toward the array center is a decayed mode with a significant attention factor [17].

Now, the proposed antenna array is introduced to a metamaterial unit cell column in the back panel to eliminate the mutual coupling between the antenna edges. In our design, we introduced the proposed unit cell parametrically, from one to four with the step of the one-unit cell, to minimize the mutual coupling altimetry. Therefore, it is observed from the evaluated S-parameters results, in Figure 10, that a significant enhancement in the antenna bandwidth is achieved after the proposed metamaterial introduction (see Figure 10a) with an increasing metamaterial unit cell number. Consequently, the antenna mutual coupling was found to decrease when increasing the unit cell number that maintains the separation distance between the antenna elements about 0.1 λ at 3.5 GHz. Therefore, we found from the proposed parametric study that the minimum S_12_ reduction is achieved when four-unit cells are introduced to the proposed array, as seen in Figure 10b. Such a reduction, it is good to mention, was found to be overall in the frequency band from 3.5 to 5.5 GHz with S_12_ below −20 dB and S_11_ ≤ −10 dB. It is concluded from these results that the proposed metamaterial array behaves as a stop-band filter to reject most of the frequencies within the frequency band of interest, as discussed later.

We next evaluated the antenna TRAC using the effective antenna gain for the optimal antenna performance with different signal phase excitations at Port 2 of 0°, 30°, 60°, and 90°. The obtained results are depicted in Figure 11a, showing a significant effect of changing the signal phase difference on the evaluated TRAC in terms of impedance bandwidth. Such an observation is attributed to sever variation in the phase change due to the effects of mutual coupling at random phase signals [9].

The antenna radiation efficiency was evaluated for both cases of with and without MTM introduction. The calculated results are presented in Figure 11b, showing a significant enhancement after introducing the proposed MTM structure. This enhancement is attributed to the fact of reducing the capacitive and inductive losses by the proposed MTM introduction [7].

### 3.2. Antenna Reconfiguration Study

The proposed antenna array was designed for reconfigurable MIMO systems based on sub-6 GHz to suit the applications of 5G networks. Therefore, it is very obvious that such systems require antenna terminals of configurable performance [19]. In our design, we introduced four LDR switches to control the antenna performance. The location of positioning the proposed LDR switches was selected to be between the meander line structure and the antenna ground plane. Such a location is considered to reflect out of phase the surface wave, at 90° phase difference, and to cancel the capacitive part inductively; this realizes the frequency shift through imaginary part consolation [20].

Our target was to control the surface current on the antenna element by controlling the real impedance at a certain point, in which most of the charge accumulation could accrue [20]. It is well known from the literature that such an accumulation of electric chargers happens at the meander lines corners [21]. Therefore, it is very wise to consider these locations for introducing the LDR switches. Nevertheless, the ground plane is a very saturated conductive surface that realizes an excellent phase reflection to obtain the imaginary part consolation for frequency reconfiguration [22]. In such a manner, each antenna element is introduced to four LDR switches to control the antenna performance in terms of frequency bandwidth and gain, as listed in the antenna truth Table 3. It was found from the study that a significant change could accrue when switching scenarios of the proposed LDR terminals change. Such a variation in the antenna response is attributed to the fact of varying the surface current on the antenna path and the inductive/capacitive behaviors to realize a significant bandwidth variation. It is good to mention that the proposed work was conducted only for these cases due to space limitations. For more details, the authors present the variation in the obtained S_11_ and S_12_ spectra in Figure 12, which shows non-remarkable variation in the antenna mutual coupling. This is because S_12_ spectra are usually a measure of coupling between the adjacent antennas that has no response to LDR-switching scenarios. This observation was discussed earlier in [22].

## 4. Results, Discussion, and Validation

Now, to validate the proposed antenna array performance experimentally, the proposed array was fabricated using a chemical wet etching process. As seen in Figure 13, the fabricated prototype was presented with maximum dimensions of 50 × 30 mm^2^. The proposed antenna S-parameters were measured using an HP vector network analyzer.

The measured antenna radiation patterns were considered at 3.5, 4, 4.5, 5, and 5.5 GHz inside an RF chamber. This process was invocated after calibrating all processes in the channel using the through calibration technique, and open, short, and 50 Ω load. The obtained result from the experimental measurements was evaluated for only two switching scenarios; when all LDR terminals are on and off only. The obtained results are shown in Figure 14 for the two considered scenarios in terms of S-parameters. An excellent agreement between measured and simulated results was found.

Below, the antenna radiation measurements are presented in Figure 15 at the frequency band of interest. The measurement reveals good agreement between the simulated and measured results. It is good to mention that the presented results of the radiation patterns are for the cases of all LDR switched on and off only. From the obtained results we found that a variation accrues in the antenna directivity due to the surface current change as discussed later. It is good to mention that the antenna measurements were obtained at theta = 0° and 90° planes, at which the gain is maximum. Additionally, the maximum E-field can be achieved at theta = 90° and the maximum H-field can be achieved at theta = 0°.

The antenna performance in terms of correlation and diversity envelopes was evaluated experimentally to be compared with the simulated results in Figure 15. The achieved results show insignificant mutual coupling between the antenna elements to realize a maximum correlation factor below 1% with a high diversity of about 99% within the frequency band of interest. This is attributed to the effect of the proposed MTM array reducing the mutual coupling significantly without an effective separation distance. For further analysis, the authors applied a study to realize the MIMO antenna performance in terms of the envelop correlation coefficient (ECC), diversity gain (DG), channel capacity loss (CCL), and total active reflection coefficient (TARC) according to the process that was explained in [23]. As seen in Figure 16, the proposed antenna system realizes a low value of ECC of about 1% with a high diversity of 10 dB over the entire frequency band of interest. This is due to the introduction of the proposed filter design to achieve a maximum mean effective gain (MEG) of about −5 dBi, while the maximum CCL value is about 0.06 bps/Hz for the entire bandwidth. The measured TRAC (at a signal phase difference of 90° only) was found to be −15 dB, which shows the capability of the proposed MTM inclusions to reduce the correlation factor between the antenna elements. It is good to mention that the evaluated ECC spectrum was obtained from the S-parameters.

To realize the effects of using the proposed MTM on the proposed antenna performance, a comparison study is summarized at the end of the discussions to focus on the role of the proposed MTM unit cell. In Table 4, the summary is listed to show the complete picture of such a discussion in terms of frequency bandwidth, gain, and coupling. The antenna array performance with and without the proposed MTM is included in Table 4 at fixed separation distances for both considered cases.

Now, to present a complete picture of the proposed antenna array performance, the authors evaluated the channel capacity (CC) and the bit error rate (BER) under the assumption of constructing an antenna array of 64 × 64 elements, theoretically only; for such an assumption, we tried to perform an analytical validation of such antenna feasibility for massive MIMO systems. The calculated CC and BER are based on measured results from the antenna elements in terms of gain, bandwidth, and mutual coupling. The channel is affected by white Gaussian noise and the transmitted power was changed from −20 to 20 dBm. In such circumstances, the maximum bit error rate was found to be about 0.15, and the channel capacity was found to reach 2 Gbps. The obtained results are presented in Figure 17.

The evaluated BER and CC were calculated using the following mathematical expressions [26]:$$BER=erfc\left(\sqrt{2E_{b}/N_{o}}\right)\;CC=BW\times \left(MN\right)\;log_{2}\left(1+SNR\right)$$
where *erfc* is the complementary error function, *E_b_* is the bit energy, *N_o_* is noise power, and *BW* is the available bandwidth.

In those calculations, a digital modulation schema based on quadrature-amplitude modulation (QAM) with an effective coverage was applied to indicate the effects of that on BER. The considered BER in this case is defined as the number of acceptable errors at the prepared tolerates. In such a case, typically, it is a number between 0.1 and 0.000001. This ratio is very much affected by the signal-to-noise ratio (SNR), which is measured in dB. It was found from the results in Figure 17a that a high SNR provides low BER, while a low SNR would increase BER. It is obvious that frequency bands with higher gain are better than those with a low gain to provide a lower chance of errors due to the noise effects. Later, the proposed antenna array CC spectra were evaluated and shown in Figure 17b. It was found that the proposed antenna array CC capacity increases from 1.5 to 4.5 Gbps with a frequency increase.

Finally, the proposed antenna array performance was compared with the published results in the literature in terms of antenna bandwidth, gain, size, reconfigurability, and substrate type. It was observed that the proposed antenna array shows a maximum-size reduction with excellent configuration technique and performance in comparison with the previous one listed in Table 5. Such a reconfiguration technique requires no biasing circuit or wiring system, while most of the old traditional techniques came up with high biasing complexity and wiring systems.

## 5. Conclusions

In this paper, the authors designed a miniaturized antenna array for MIMO applications. The proposed antenna parameters were optimized to fit the applications of sub-6 5G networks in terms of mutual coupling reduction, bandwidth enhancements, and gain increase. Therefore, the authors applied in the proposed design several techniques to realize a highly miniaturized antenna array with excellent reconfiguration mechanisms. For this, a monopole fundamental antenna element is conjugated with a meander line and interdigital capacitor to increase the antenna radiation efficiency with low surface wave retardation. In such a manner, fitting the proposed antenna element in a single array could be an excellent addition to MIMO systems by minimizing the separation distance between them using a metamaterial structure. The proposed metamaterial was found to realize a significant mutual reduction of almost −20 dB over the frequency band from 3 up to 5.5 GHz with a separation distance of 0.08 λ. This antenna array was tested with different configuration scenarios by adding four LDR switches to each antenna element through the meander line and the ground plane. We realized that such a reconfiguration process takes advantage of low complexity without including wiring or biasing systems. It was found that when exposed to a massive MIMO environment in the proposed antenna array, acceptable BER and CC can be achieved after making the antenna size theoretically 64×64 elements. It is concluded from this study that the proposed antenna array is a very suitable candidate for 5G massive MIMO systems at the sub-6 GHz frequency band. Finally, for validation, an experimental study was applied to the proposed array to test the antenna performance in terms of S-parameters and radiation patterns.

## Figures and Tables

**Figure 1 sensors-23-05871-f001:**
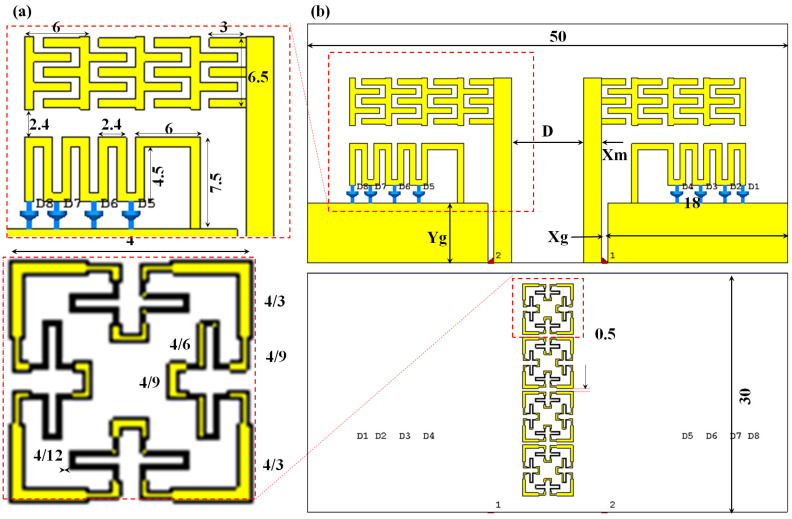
Antenna geometrical details in mm scale: (**a**) front view and (**b**) back view.

**Figure 2 sensors-23-05871-f002:**
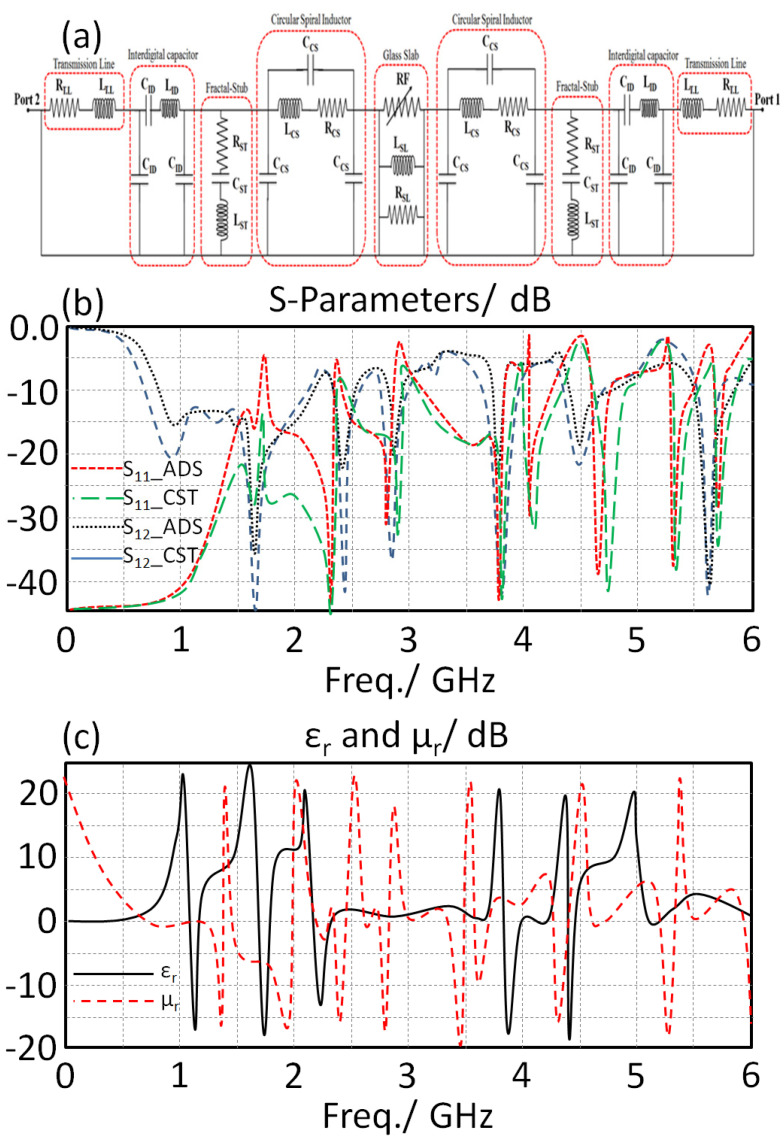
The proposed MTM structure circuit model: (**a**) equivalent circuit, (**b**) S-parameters results, and (**c**) retrieved electromagnetic properties in terms of ε_r_ and µ_r_.

**Figure 3 sensors-23-05871-f003:**
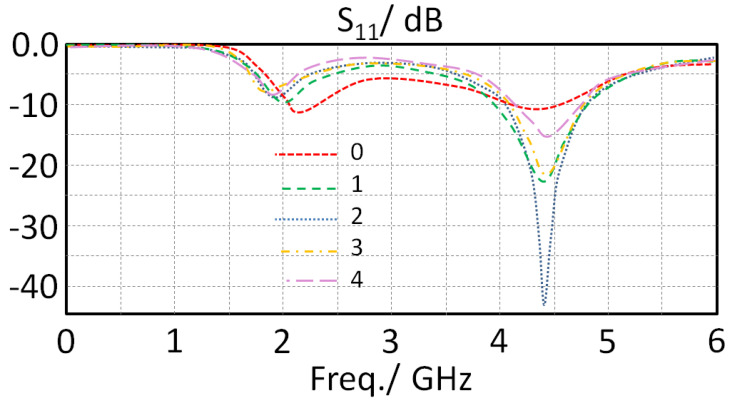
The obtained S_11_ spectra variation for the proposed antenna with changing Xg.

**Figure 4 sensors-23-05871-f004:**
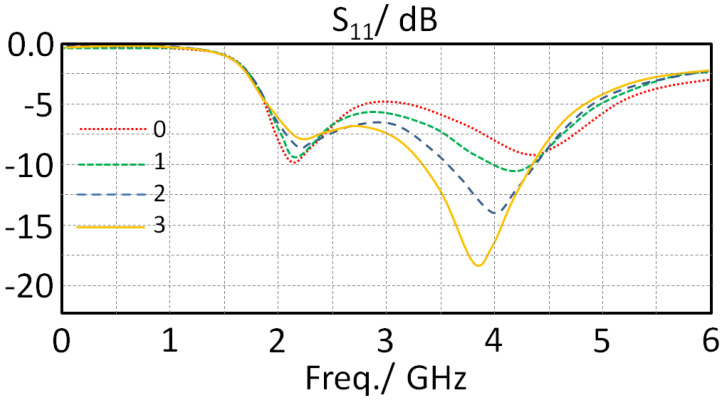
The obtained S_11_ spectra variation for the proposed antenna with changing Yg.

**Figure 5 sensors-23-05871-f005:**
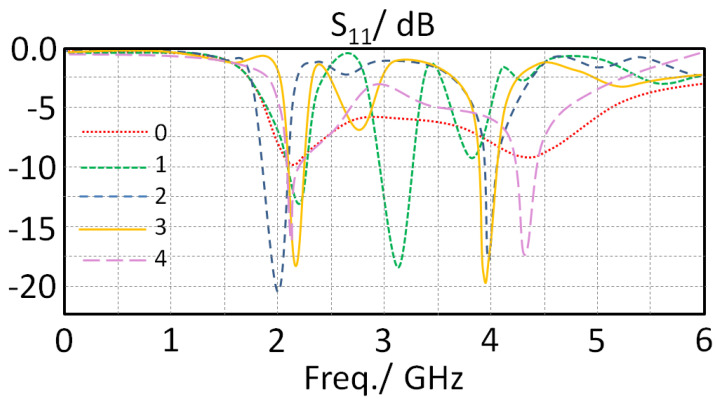
The obtained S_11_ spectra variation for the proposed antenna with changing Xm.

**Figure 6 sensors-23-05871-f006:**
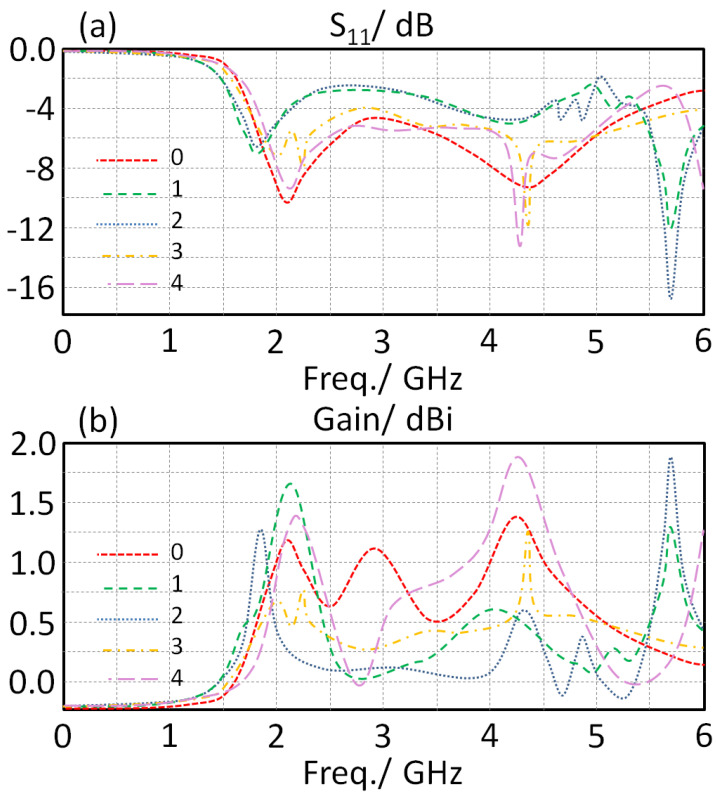
Obtained antenna performance variation with changing IDC iteration: (**a**) S_11_ and (**b**) gain spectra.

**Figure 7 sensors-23-05871-f007:**
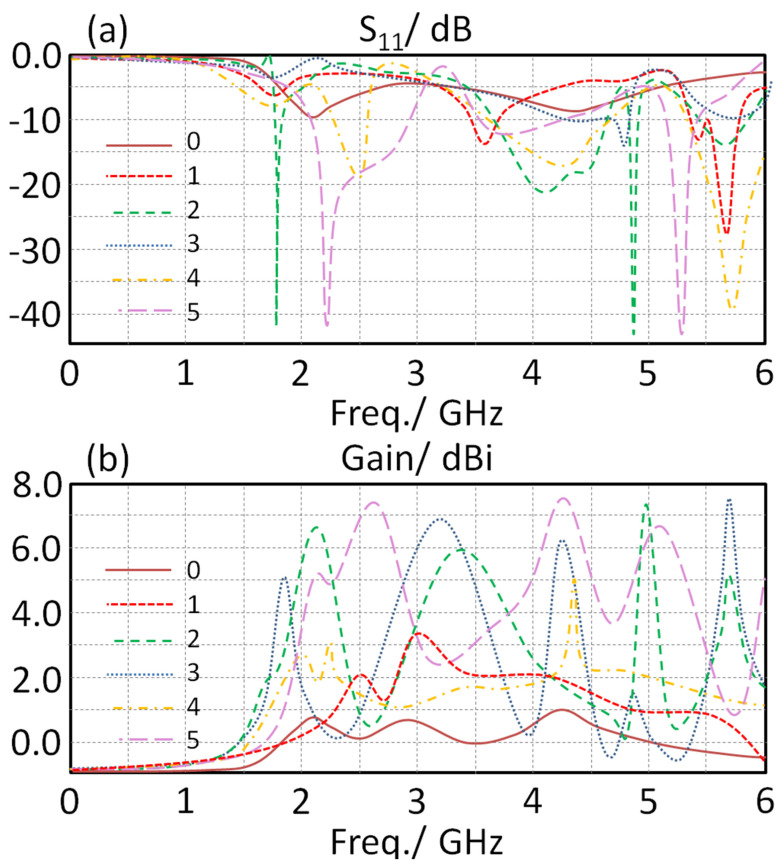
The obtained S_11_ and gain spectra for the proposed antenna with changing IDC iteration: (**a**) S11 and (**b**) gain spectra..

**Figure 8 sensors-23-05871-f008:**
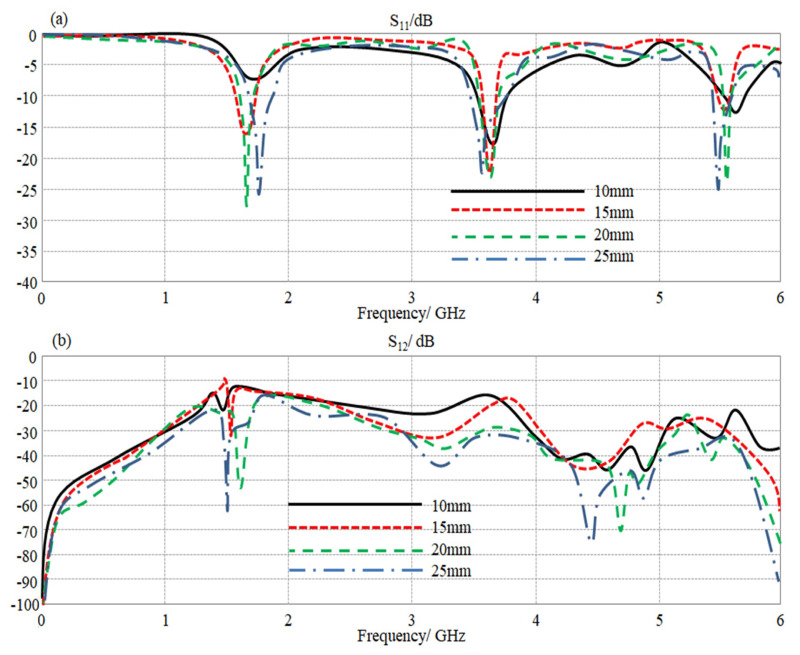
The obtained S-parameters spectra for the proposed antenna array with changing D: (**a**) S_11_ and (**b**) S_12_ spectra.

**Figure 9 sensors-23-05871-f009:**
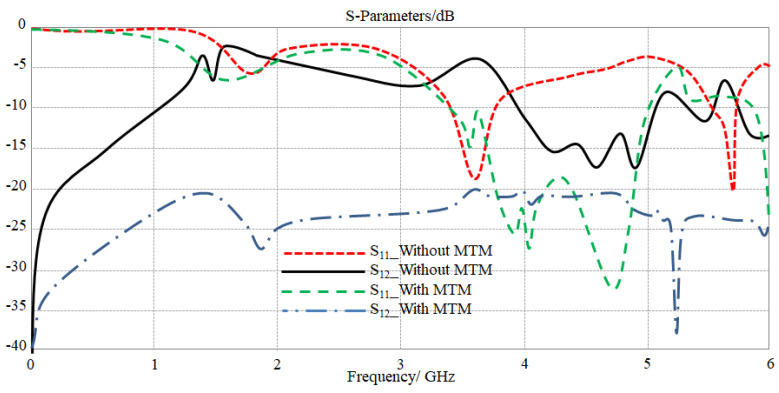
The obtained S-parameters spectra for the proposed antenna array with and without MTM defects.

**Figure 10 sensors-23-05871-f010:**
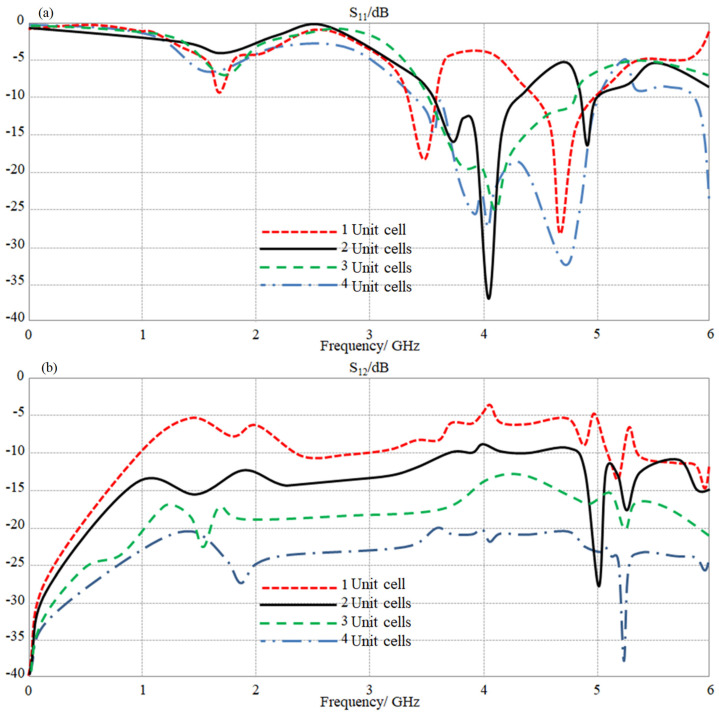
The obtained S-parameters spectra for the proposed antenna array with changing MTM defects number: (**a**) S_11_ and (**b**) S_12_ spectra.

**Figure 11 sensors-23-05871-f011:**
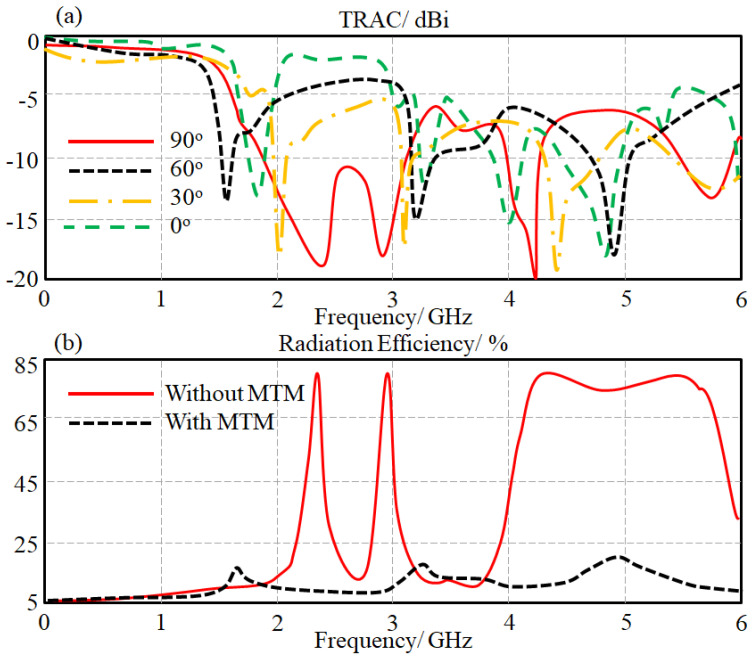
Array performance in terms of (**a**) the obtained TRAC spectra at different signal phase excitations and (**b**) radiation efficiency spectra.

**Figure 12 sensors-23-05871-f012:**
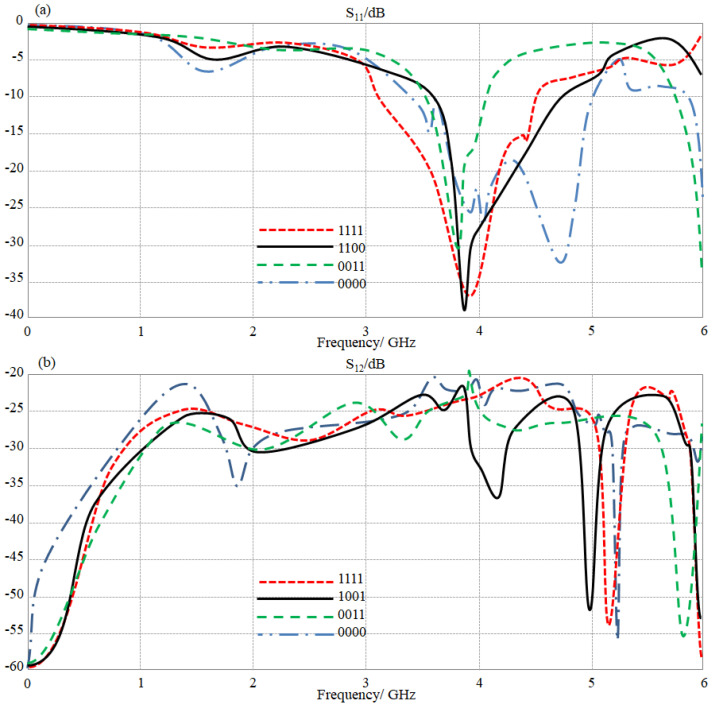
The obtained S-parameters spectra for the proposed antenna array with changing switching scenarios: (**a**) S_11_ and (**b**) S_12_ spectra.

**Figure 13 sensors-23-05871-f013:**
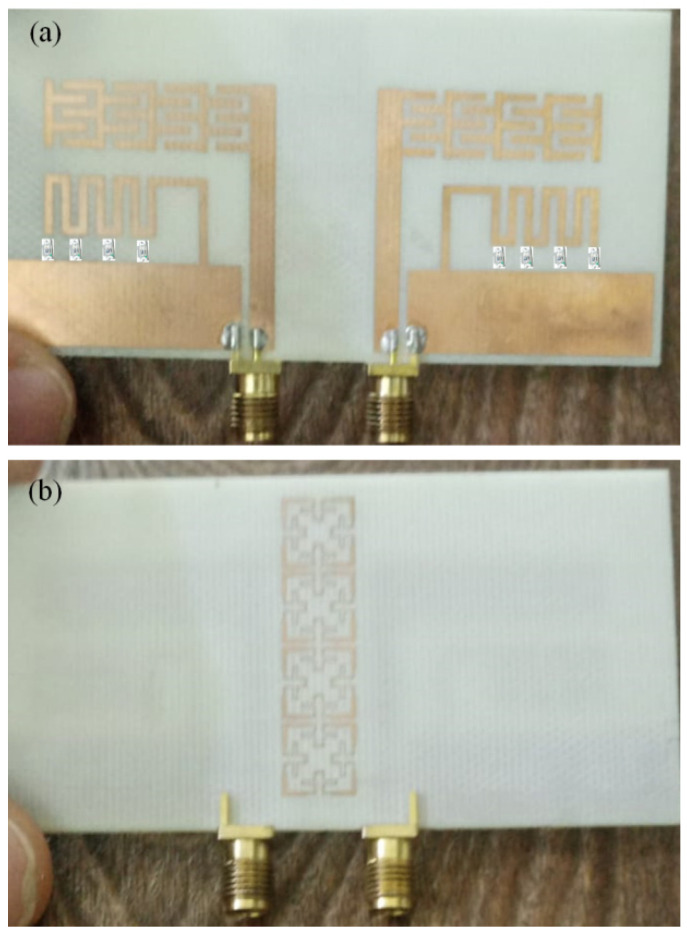
The proposed prototype: (**a**) front view and (**b**) back view.

**Figure 14 sensors-23-05871-f014:**
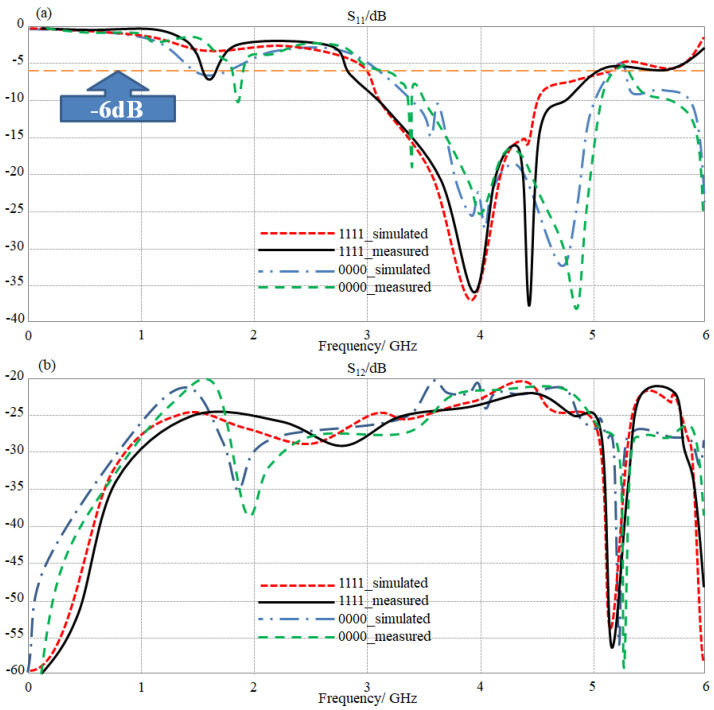
The measured S-parameters spectra for the proposed antenna array with changing switching scenarios: (**a**) S_11_ and (**b**) S_12_ spectra.

**Figure 15 sensors-23-05871-f015:**
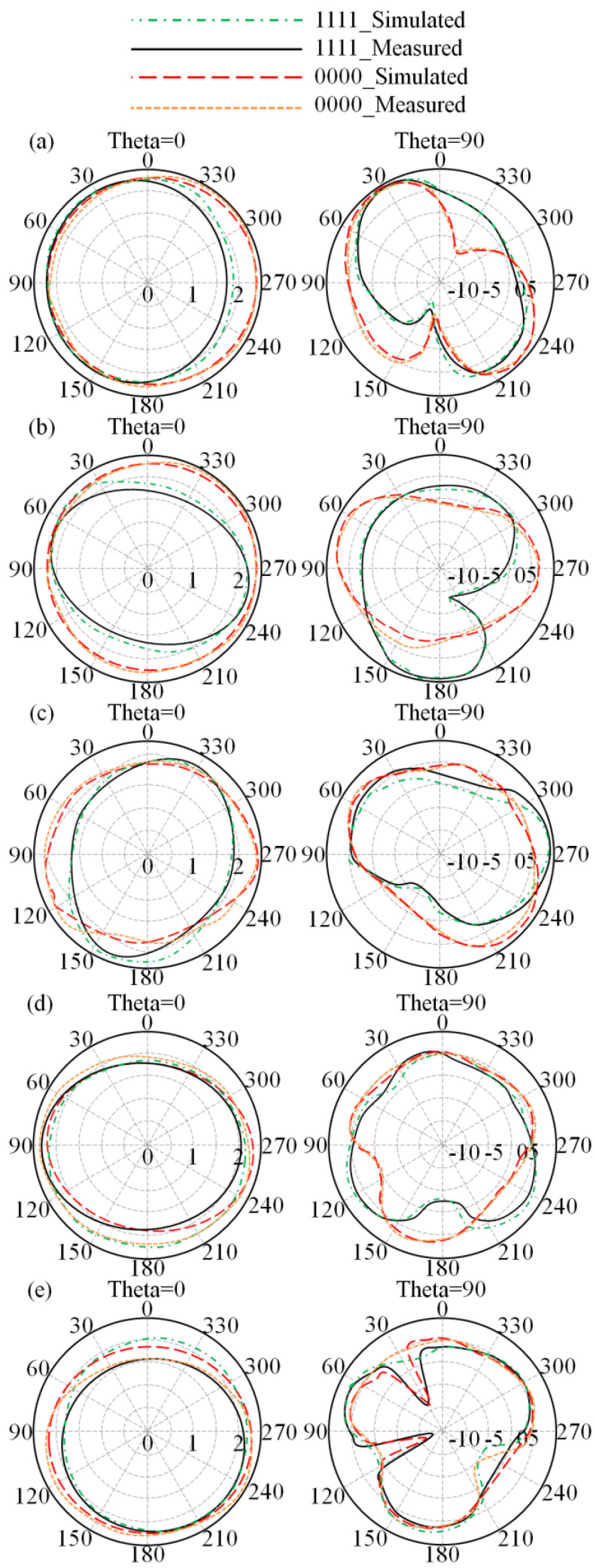
Measured antenna radiation patterns at different frequency bands: (**a**) 3.5 GHz, (**b**) 4 GHz, (**c**) 4.5 GHz, (**d**) 5 GHz, and (**e**) 5.5 GHz.

**Figure 16 sensors-23-05871-f016:**
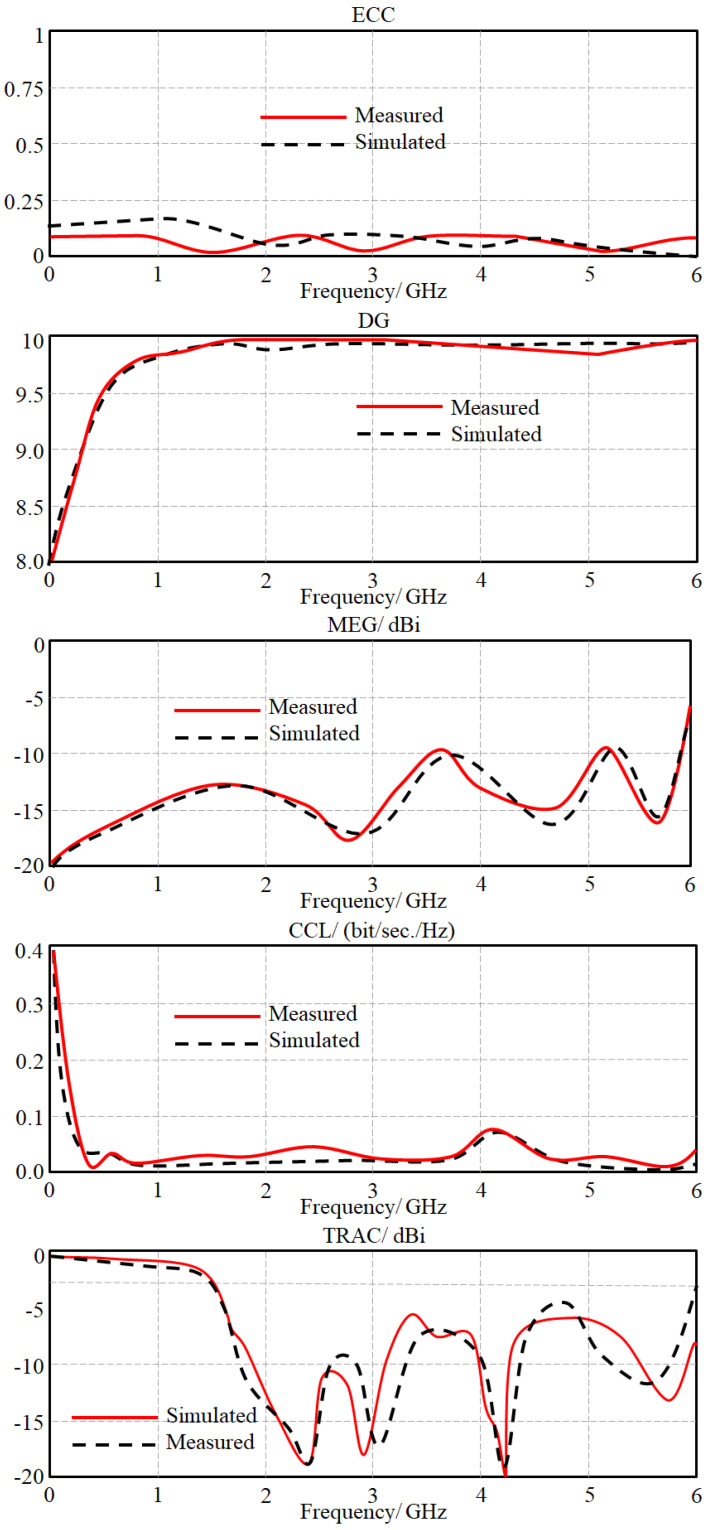
Antenna array performance in terms of ECC, DG, MEG, CCL, and TRAC.

**Figure 17 sensors-23-05871-f017:**
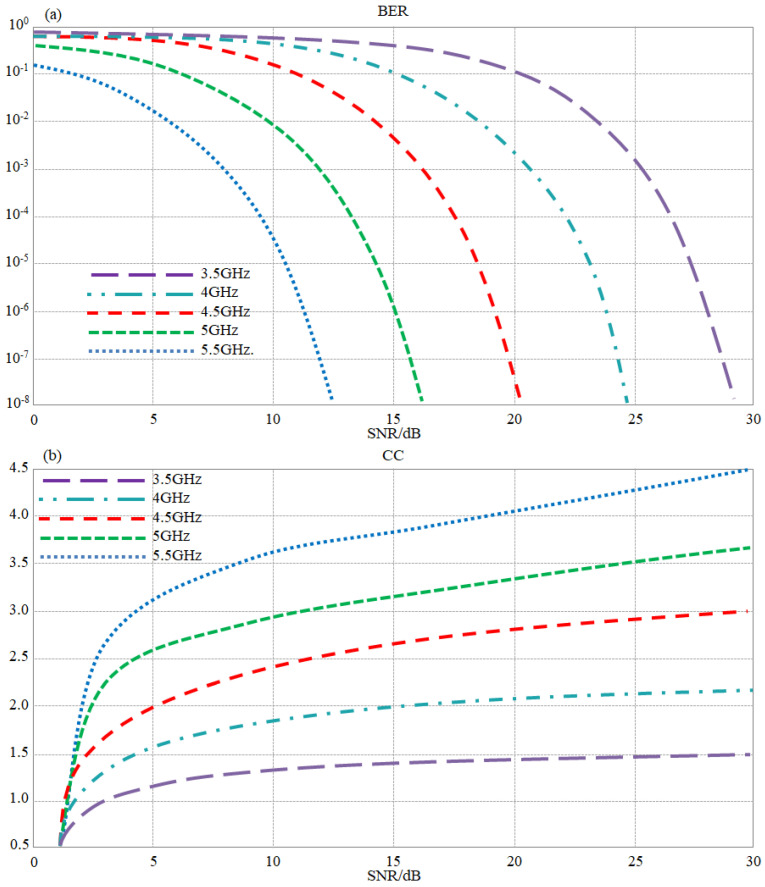
Channel performance calculations at different frequency bands: (**a**) BER and (**b**) CC.

**Table 1 sensors-23-05871-t001:** Antenna geometrical details.

Symbol	Value	Abbreviation
Xg	3	Separation distance
Yg	3	Ground plane length
Xm	4	Monopole width
D	12	Separation distance between antenna elements

**Table 2 sensors-23-05871-t002:** Equivalent circuit model lumped elements.

Element	Value
Resistance of the left hand (RLH)	12.2 Ω
Resistance of the right hand (RRH)	50 Ω
Conductance of the left hand (GLH)	0.11 S
Conductance of the right hand (GRH)	4.55 S
Capacitance of the left hand (CLH)	1.17 pF
Capacitance of the right hand (CRH)	3.11 pF
Inductance of the left hand (LLH)	3.24 nH
Inductance of the right hand (LRH)	2.34 nH

**Table 3 sensors-23-05871-t003:** Antenna performance with different switching scenarios.

Switching Scenario	Frequency/GHz	Gain/dBi
0000	3–5.5	4.5
0011	3.5–4.7	5.1
1100	3.4–4.1	6.2
1111	3.1–4.6	8.1

**Table 4 sensors-23-05871-t004:** The proposed antenna performance with and without MTM introduction at fixed antenna sizes and separation distances for both cases.

Case	BW/GHz	Gain/dBi	Efficiency	DG	ECC	Coupling/dB
Without MTM	3.2–4.65	1.1	20%	5.3	0.24	−3
With MTM	3–5.5	4.5	83%	10	0.01	−20

**Table 5 sensors-23-05871-t005:** The proposed antenna performance in comparison to published results.

Ref.	Size	Ports	BW/GHz	Gain/dBi	Coup/dB	DG	ECC	D	Rec.
[18]	130 × 100	8	5.1–5.9	2.1	−15	---	0.05	λ/1.9	Not
[19]	150 × 75	4	2.45,5.8	2.5	−14	9.8	0.06	λ/2	Not
[20]	136 × 60	8	5.1–5.9	1.9	−10	9.3	0.09	λ/2	Diode
[21]	150 × 75	12	4.8–5.1	2.6	−12	9.7	---	λ/2.1	Not
[22]	150 × 80	8	5.1–5.9	2.2	−10	9.2	0.11	λ/2.3	Not
[23]	133 × 133	4	2.25	2.5	−15	9.3	0.10	λ/2.1	Diode
[24]	160 × 70	2	4.42	5.2	−30	10	0.06	λ/1.4	Not
[25]	30 × 40	4	2–3, 3.4–3.9, 4.4–5.2	6.3	−20	10	0.01	λ/15	Not
This work	30 × 50	2	3–5.5	4.5	−20	10	0.01	0.08λ	LDR

## Data Availability

Not applicable.

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
