# Peer review of "Miniaturized Antenna Array-Based Novel Metamaterial Technology for Reconfigurable MIMO Systems"

_sensors, 2023, doi:10.3390/s23135871_

Round 1
Reviewer 1 Report
In the Manuscript entitled ‘Miniaturized Antenna Array based Novel Metamaterial Technology for Reconfigurable MIMO Systems’, Author’s have presented a metamaterial-based miniaturization technique for MIMO antenna. To improve the quality of the manuscript, bellow points are highlighted to mention/clarify in the manuscript.
1. In Figure 1, the Metamaterial (MTM) loaded antenna has been presented. However, the parameter extraction (like permittivity, permeability, and refractive index) of MTM is required to include in the manuscript and also which particular property has been used to attain miniaturization is not clear. Clarify that in manuscript.
2. Improve the quality of the Figure 1.
3. In caption of Figure 2(b), S12 result is written, but in graph it is S21. Rectify this.
4. Improve the quality of the Figure 3 to 7.
5. By using MTM, which parameters of the antenna got enhanced in not clear. To clarify this, include a comparative study Table between the antenna performance with and without MTM including antenna frequency, gain, directivity, efficiency, coupling and miniaturization etc.
6. In Figure 14, all the plots for Theta 900 got degraded/tilted compared to Theta 00 include the explanation.
7. It seems the paper is written in casual way, check the manuscript thoroughly and revised it.

Check the manuscript thoroughly and revised it.
Author Response
Reviewer_1
- In Figure 1, the Metamaterial (MTM) loaded antenna has been presented. However, the parameter extraction (like permittivity, permeability, and refractive index) of MTM is required to include in the manuscript and also which particular property has been used to attain miniaturization is not clear. Clarify that in manuscript.
Answer: This is considered in the revised version.
- Improve the quality of the Figure 1.
Answer: All figures are fixed.
- In caption of Figure 2(b), S12 result is written, but in graph it is S21. Rectify this.
Answer: this issue is fixed in all figures are fixed.
- Improve the quality of the Figure 3 to 7.
Answer: All figures are fixed.
- By using MTM, which parameters of the antenna got enhanced in not clear. To clarify this, include a comparative study Table between the antenna performance with and without MTM including antenna frequency, gain, directivity, efficiency, coupling and miniaturization etc.
Answer: This is considered in the revised version.
- In Figure 14, all the plots for Theta 900 got degraded/tilted compared to Theta 00 include the explanation.
Answer: This is considered in the revised version.
- It seems the paper is written in casual way, check the manuscript thoroughly and revised it.
Answer: This is considered in the revised version.
Reviewer 2 Report
In this paper, authors have reported MIMO antenna and tried to reduce the mutual coupling between the radiating elements by using Metamaterial unit cells in between the patches at opposite surface of dielectric. I have gone through this paper and recommend following suggestions:
1. Please correct the English grammar throughout manuscript, there are many grammatical errors.
2. In introduction, please mention the acronym for CRLH
3. In many places starting from abstract, statements are in inverted comma e.g. “multiple inputs multiple outputs (MIMO) application” , I think inverted comma are not required and it can be removed.
4. In table, symbols Xg, Yg, Xm and D should be mentioned in Figure 1.
5. IDC and CSI acronyms are not mentioned in the text.
6. In Fig. 2, please check whether it is S12 or S21 graph.
7. The results as shown in Fig 5 is not acceptable, please plot the graph of S11 at acceptable/optimised value of Xm, otherwise this curve is useless.
8. Fig 7 indicates the insufficient information as IDC iteration values are not mentioned in the curve.
9. In Fig 10, put the legends as no of unit cell.
10. Antenna efficiency curve is missing for the MIMO antenna.
11. In Fig 15, please correct the y-axis parameters with units.
12. Which parameters (S-parameters or radiation field) are used to calculate the ECC of MIMO antenna, please mention.
13. One major issue I have observed in the TARC value of the MIMO antenna as shown in Fig. 15. Why the TARC value is so much different as compared to the S11 value of the MIMO design? In principle, TARC represent the variation of S11 values at different phase angle of signal. In my view this result is unacceptable. Please correct it and plot TARC value for different phase angles of input signal.
14. Authors must report the mathematical expression for the calculation of channel capacity and the bit error rate
15. References are not up to the standard, as most are from conferences. Please add some references from good journals.
English Grammar needs to be improved in the manuscript.
Author Response
Reviewer_2
- Please correct the English grammar throughout manuscript, there are many grammatical errors.
Answer: The paper is revised well. This is considered in the revised version.
- In introduction, please mention the acronym for CRLH
Answer: This is considered in the revised version.
- In many places starting from abstract, statements are in inverted comma e.g. “multiple inputs multiple outputs (MIMO) application” , I think inverted comma are not required and it can be removed.
Answer: This is considered in the revised version.
- In table, symbols Xg, Yg, Xm and D should be mentioned in Figure 1.
Answer: This is considered in the revised version.
- IDC and CSI acronyms are not mentioned in the text.
Answer: This is considered in the revised version.
- In Fig. 2, please check whether it is S12 or S21 graph.
Answer: This is considered in the revised version.
- The results as shown in Fig 5 is not acceptable, please plot the graph of S11 at acceptable/optimised value of Xm, otherwise this curve is useless.
- Fig 7 indicates the insufficient information as IDC iteration values are not mentioned in the curve.
Answer: This is considered in the revised version.
- In Fig 10, put the legends as no of unit cell.
Answer: This is considered in the revised version.
- Antenna efficiency curve is missing for the MIMO antenna.
Answer: This is considered in the revised version.
- In Fig 15, please correct the y-axis parameters with units.
Answer: This is considered in the revised version.
- Which parameters (S-parameters or radiation field) are used to calculate the ECC of MIMO antenna, please mention.
Answer: This is considered in the revised version.
- One major issue I have observed in the TARC value of the MIMO antenna as shown in Fig. 15. Why the TARC value is so much different as compared to the S11 value of the MIMO design? In principle, TARC represent the variation of S11 values at different phase angle of signal. In my view this result is unacceptable. Please correct it and of input signal.
Answer: This is considered in the revised version in Figure 11.
- Authors must report the mathematical expression for the calculation of channel capacity and the bit error rate
Answer: This is considered in the revised version.
- References are not up to the standard, as most are from conferences. Please add some references from good journals.
Answer: Many references are changed in the revised version. If there is any further reference please we hope to suggest.
Round 2
Reviewer 2 Report
The authors have incorporated the suggestions. However, I recommend the improvement in the English language throughout the paper.
Needs some improvement